# Combination of JAKi and HDACi Exerts Antiangiogenic Potential in Cutaneous T-Cell Lymphoma

**DOI:** 10.3390/cancers16183176

**Published:** 2024-09-17

**Authors:** Fani Karagianni, Christina Piperi, Sara Valero-Diaz, Camilla Amato, Jose Pedro Vaque, Berta Casar, Evangelia Papadavid

**Affiliations:** 1National Center of Rare Diseases-Cutaneous Lymphoma, Second Department of Dermatology and Venereal Diseases, Attikon University General Hospital, National and Kapodistrian University of Athens, 12462 Athens, Greece; karagiannifani@gmail.com (F.K.); cpiperi@med.uoa.gr (C.P.); 2Department of Biological Chemistry, National and Kapodistrian University of Athens, 11527 Athens, Greece; 3Instituto de Biomedicina y Biotecnología de Cantabria, Consejo Superior de Investigaciones Científicas (CSIC)-Universidad de Cantabria, Santander 39011, Spain; svd413@alumnos.unican.es (S.V.-D.); camilla.amato@studenti.unimi.it (C.A.); 4Department of Medical Biotechnology and Molecular Medicine, Università degli Studi di Milano, 20122 Milan, Italy; jose.vaque@unican.es; 5Molecular Biology Department, Universidad de Cantabria-Instituto de Investigación Marqués de Valdecilla, IDIVAL, 39011 Santander, Spain; 6Centro de Investigación Biomédica en Red de Cáncer (CIBERONC), Instituto de Salud Carlos III, 28220 Madrid, Spain

**Keywords:** ruxolitinib, resminostat, angiogenesis, CTCL, chick embryo model

## Abstract

**Simple Summary:**

The combination of JAK/HDAC inhibitors shows beneficial effects in hematological malignancies and a promising therapeutic potential in CTCL. Recent in vitro and in vivo data have shown that the combination of Resminostat (HDACi) with Ruxolitinib (Jaki) acts synergistically and inhibits tumorigenesis and metastasis in CTCL experimental models. Angiogenesis is the most critical process for tumor growth and metastasis and plays an essential role in cutaneous lymphoma development and progression. The vascular microenvironment of lymphomas accelerates angiogenesis via the paracrine activity of tumor-derived mediators. However, few studies have investigated the role of either ruxolitinib or resminostat in the angiogenesis of hematological malignancies and solid tumors. Although precise mechanisms of angiogenesis in CTCL remain unclear, the chick embryo chorioallantoic membrane (CAM) has been used, among other in vivo models, to implant several tumor types as well as malignant cell lines to study their growth rate, angiogenic potential, and metastatic capability. The aim of this study was to investigate the effects of combinational therapies of JAKi/HDACi in CTCL angiogenesis by using a CTCL-specific chick embryo model.

**Abstract:**

Angiogenesis plays a pivotal role in the growth and metastasis of tumors, including the development and progression of cutaneous lymphomas. The chick embryo CAM model has been utilized in various studies to assess the growth rate, angiogenic potential, and metastatic capability of different tumor types and malignant cell lines. However, the precise mechanisms of angiogenesis in CTCL and the influence of Ruxolitinib or Resminostat on angiogenesis in hematological malignancies and solid tumors are not well understood. Recent in vitro and in vivo data have demonstrated the synergistic inhibition of tumorigenesis and metastasis in experimental models of CTCL when using the combination of Resminostat (HDACi) with Ruxolitinib (JAKi). The present work aims to elucidate the effects of this combination on the tumor microenvironment’s vascular components. We investigated the effects of Ruxolitinib (JAKi) in combination with Resminostat (HDACi) treatment in transendothelial migration of CTCL cells (10^6^ MyLa and SeAx) by using a transwell-based transendothelial migration assay and tumor angiogenesis in vivo. We used the CTCL chick embryo CAM model with xenografted tumors derived from implanted MyLa and SeAx cells and administered topically 15 μM ruxolitinib and 5 μM Resminostat every two days during a 5-day period. JAKi and HDACi inhibited CTCL cell transendothelial migration by 75% and 82% (*p* < 0.05) in both CTCL engrafted cells (MyLa and SeAx, respectively) compared to the untreated group. Moreover, the combination of ruxolitinib with resminostat blocked angiogenesis by significantly reducing the number of blood vessel formation by 49% and 34% in both MyLa and SeAx, respectively (*p* < 0.05), indicating that the proposed combination exerted significant anti-angiogenic effects in the CAM CTCL model. Overall, these data provide valuable insights into potential therapeutic strategies targeting angiogenesis in CTCL, paving the way for more effective treatment approaches in the future.

## 1. Introduction

Mycosis fungoides (MFs), the most common type of Cutaneous T-cell lymphoma (CTCL), accounts for ~50% of all primary cutaneous lymphomas, followed by Sézary syndrome. CTCL is characterized by the recruitment of malignant T-cell clones into the skin. In 2018, the EORTC cutaneous lymphoma task force proposed uniform classification and prognostic methods for MF and Sézary syndrome using flow cytometry [1]. The initiation and progression of CTCL involve various molecular mechanisms such as dysregulated genes, proteins, signaling pathways, and interactions within the tumor microenvironment [2]. Aberrant activation of Janus kinase (JAK) and signal transducer and activator of transcription (STAT) signaling has been implicated in CTCL. Mutations in JAK/STAT pathway components or dysregulated cytokines can contribute to increased proliferation, survival, and evasion of the immune system by malignant T cells [3]. Moreover, cytokine signaling such as IL-2, IL-4, IL-15, IL-21, and IL-31 can promote T-cell proliferation, survival, and migration, contributing to CTCL pathogenesis. Furthermore, alteration in epigenetic regulation, including DNA methylation, histone modifications, and microRNA expression, can affect gene expression patterns in CTCL. Epigenetic changes often contribute to the dysregulation of key genes involved in cell cycle control, apoptosis, and immune response [3]. It is also important to note that CTCL progression is influenced by interactions within the tumor microenvironment. Crosstalk between malignant T cells and surrounding stromal cells, immune cells, and the extracellular matrix components contribute to disease progression by modulating the immune response, promoting angiogenesis, and providing survival signals to malignant T cells. Last but not least, the development of blood and lymphatic vessels was associated with the progression of CTCL [3]. 

Angiogenesis constitutes an important player in pathologic conditions in which uncontrolled release of angiogenic growth factors and alterations of the production of natural angiogenic inhibitors take place [4]. The vascular niche plays a crucial role in tumor growth, producing growth factors like VEGF and FGFb [5]. Moreover, the vascular microenvironment in lymphomas, including CTCLs, plays a significant role in promoting neoangiogenesis, facilitating disease progression, and metastasis. Several factors released by tumoral cells contribute to this process, including members of the Vascular Endothelial Growth Factor (VEGF) family, basic Fibroblast Growth Factor (bFGF), and Placental Growth Factor (PlGF) [3]. The process of tumor angiogenesis is characterized as a dynamic and pivotal mechanism contributing significantly to both the initiation and progression of tumors. Tumor cells actively engage in the synthesis and secretion of angiogenic factors. These factors, in turn, exert their influence by stimulating vascular permeability and promoting endothelial cell proliferation. This intricate interplay plays a fundamental and decisive role in the advancement and evolution of malignant breast tumors [6]. VEGF represents the main angiogenic target for the development of antiangiogenic therapies in cancer patients [7,8,9,10,11]. An association between angiogenesis and prognostics of MF has been well established [9,10,11,12], and repeated attempts have been made to target it therapeutically [9,13].

Previous studies have demonstrated the effect of ruxolitinib in other hematological malignancies, showing inhibition of VEGF, HIF-1α expression, and angiogenesis of HEL leukemia cells [13], and a dose-dependent inhibition of cell proliferation with concurrent activation of apoptosis, a marked and rapid inhibition of STAT activation, and inhibition in DNA synthesis in CTCL cells [14]. 

On the other hand, the multifaceted effects of HDAC (Histone Deacetylase) inhibitors on cancer cells make them potential candidates for therapeutic interventions in hematological malignancies and CTCL. However, it is important to emphasize that although research in this field is promising, clinical applications may vary, and further studies and clinical trials are necessary to establish the safety and efficacy of HDAC inhibitors in specific cancer types [15,16,17,18,19,20]. Therefore, HDAC inhibitors have shown promising therapeutic effects in the treatment of CTCLs. The primary action of HDAC inhibitors involves modification of gene expression through their effect on histone acetylation, thereby influencing various cellular processes such as proliferation, differentiation, and apoptosis of cancer cells.

The use of HDAC inhibitors in CTCL gained attention when a phase 1 trial in 2001 investigated the effects of romidepsin on various cancers. Three patients with CTCL enrolled in the trial presented partial remission, while a patient with Peripheral T-cell Lymphoma (PTCL) experienced complete remission. These initial results provided early evidence of the potential efficacy of HDAC inhibitors in treating CTCL [18]. Vorinostat is one of the HDAC inhibitors that received approval for the treatment of CTCL. It was among the first HDAC inhibitors to be approved specifically for this indication. It works by inhibiting HDAC enzymes, leading to the accumulation of acetylated histones and regulation of gene expression, which can ultimately impact cancer cell growth and survival. The approval of vorinostat for CTCL was based on clinical studies demonstrating its effectiveness in inducing responses in patients with refractory or relapsed CTCL [18]. Clinical trials evaluating vorinostat and other HDAC inhibitors have shown their ability to induce objective responses, including complete or partial remissions, and improve symptoms in CTCL patients who have failed other treatments. Apart from vorinostat and romidepsin, other HDAC inhibitors such as belinostat and panobinostat have been investigated in clinical trials for CTCL and other hematological malignancies, showing varying degrees of efficacy [18].

The use of HDAC inhibitors represents an important therapeutic approach in the management of CTCL, offering a targeted treatment option that can provide clinical benefits for patients who have limited treatment options. Ongoing research continues to explore the efficacy of HDAC inhibitors, either alone or in combination with other therapies, aiming to optimize their use and improve outcomes for individuals with CTCL.

The combination of ruxolitinib with HDAC inhibitors has been further studied in a hematological malignancy of myelofibrosis [19]. Few clinical trials have been conducted with a combination of ruxolitinib and panobinostat, indicating a safety tolerance profile with adverse events such as anemia and thrombocytopenia [18,19,20,21,22,23]. On the contrary, the combination of ruxolitinib with another panHDACi in myelofibrosis showed poor efficiency and tolerance [22]. Even though there is great scientific interest in the particular combination of JAKi and HDACi, there are few studies investigating the effects of the combination of JAKi/HDACi in CTCL. 

Currently, there is limited research on the antiangiogenic effect of ruxolitinib [24] and HDACi (romidepsin) [25,26,27,28] in CTCL, with no available study investigating specifically the role of resminostat in angiogenesis of CTCL. 

Moreover, our current in vitro data indicate the significance of this combinational therapy in CTCL cell lines, affecting cell proliferation and apoptosis [17].These data were further validated on a chick embryo CTCL model showing the antitumor effects of the combinational therapy in cell proliferation, apoptosis, tumor size, and metastasis [25].

The present study is a continuation of previous investigations, substantiating significantly the distinct antitumor efficacy of the synergistic application of the JAK2 inhibitor ruxolitinib and the HDAC inhibitor resminostat. We provide evidence of their effectiveness in inhibiting transendothelial migration and angiogenesis of CTCL cells engrafted within a chick embryo model simulating CTCL. These data are of particular importance since both JAK2 inhibitors and HDAC inhibitors when administered as monotherapies, exhibit only partial effectiveness in the context of hematological malignancies.

## 2. Materials and Methods

### 2.1. Cell Lines and Culturing 

The human CTCL cell lines, MyLa and SeAx, were kindly provided by Dr. Michel Laurence (Skin Research Center Service de Dermatologie Hôpital Saint-Louis, INSERM, Paris, France), which were already tested for any contamination and authenticated. Both cell lines were cultured in RPMI 1640, supplemented with 10% fetal bovine serum and 1% penicillin/streptomycin, at 37 ℃ in a humidified atmosphere with 5% CO_2_, for 24 h. 

### 2.2. Drugs Tested 

The HDAC inhibitor was kindly provided by 4SC AG (Planegg-Martinsried, Upper Bavaria, Germany), and the JAK inhibitor was given by Novartis Incyte (Basel, Switzerland). Both inhibitors were dissolved in DMSO according to the manufacturers’ instructions. Untreated cells or vehicle-containing controls were treated with 0.1% DMSO for all the experiments during the indicated times. Inhibitors were used at 15 μΜ Ruxolitinib and 10 μΜ Resminostat.

### 2.3. Transendothelial Migration Transwell Assay 

HVEC Endothelial cells (1 × 10^5^) were placed in 0.1 mL endothelial culture medium on the insert undersurfaces of 6.5 mm-Transwells with 8 μm pores (Corning Inc., MA, USA). Endothelial cells were grown to confluence. Pre-labeled CTCL cells were pre-incubated with corresponding vehicles or inhibitors for 24 h, 5 × 10^4^ cells in 0.1 mL SF-DMEM were placed into individual inserts. The lower chamber was filled with DMEM-5% FBS. After 24 h, transmigrated cell imaging and quantification took place using fluorescence microscopy as a percentage of green fluorescent cells among total cells.

### 2.4. Angiogenesis CAM Assay 

The CAM angiogenesis assay was performed as previously described [26]. Briefly, type I rat-tail collagen (BD Biosciences, NJ, USA) was neutralized and prepared at a final concentration of 2 mg/mL. Cells were incorporated into the collagen mixture at a final concentration of 0.5 × 10^6^ per ml of collagen solution. Where indicated, 15 μΜ Ruxolitinib and 10 μΜ Resminostat were incorporated for 72 h into the collagen mixture. A total of 20 μL of the final collagen mixture were polymerized between the two-nylon gridded meshes to form an onplant. Four to six onplants were placed onto the CAMs of shell-less day 10 embryos developing ex ovo (3 to 5 embryos per variable). Microtumor size was analyzed after 72 h by fluorescence microscopy using FIJI software. Angiogenic vessels were scored above the upper mesh after 72 h, and an angiogenic index (number of grids with newly formed blood vessels over the total number of grids scored) was calculated for each onplant using the IKOSA^®^ CAM Assay Application. All experiments were performed at least twice. Hemoglobin concentration in the onplant lysates was determined using Hemoglobin Assay Kit Heme (Merck-Sigma-Aldrich, MO, USA), according to the manufacturer’s instructions. 

### 2.5. Expression of Angiogenesis-Related Genes

To detect the expression of angiogenic factors in CTCL tumor cells, primers for human VEGF-A and VEGF-B sequences (Sigma, MO, USA) were utilized. For tumor RNA extraction and quantification, frozen tumors were minced on dry ice, and RNA was extracted using the RNeasy Mini Kit (Qiagen, Hilden, Germany), following the manufacturer’s instructions. cDNA was synthesized using iScrip Reverse Transcription Supermix (BioRad, Hercules, CA, USA). After resuspension, nucleic acid contents were quantified using NanoDropTM 2000c (Thermo Fisher, MA, USA). Twenty ng of DNA were used for RT PCR following manufacturer instructions (PowerUp SYBR Green Master Mix, Thermo Fisher, MA, USA). PCR conditions were the following: 4 min/95 °C followed by 40 cycles of 30 s at 95 °C to denature DNA; 60 °C/30 s for primer annealing and 30 s at 72 °C for amplification. Primers used were: 

Human Forward VEGF-B: 5′ GATCCAGTACCCGAGCAGTCA 3′ 

Human Reverse VEGF-B: 5′ TGGCTTCACAGCACTCTCCTT 3′

Human Forward VEGF-A: 5′-GCA CCC ATG GCA GAA GG 3′ 

Human Reverse VEGF-A: 5′ CTC GAT TGG ATG GCA GTA GCT 3′ 

Human GAPDH Forward: 5′ GAAGGTGAAGGTCGGAGTC 3′ 

Human GAPDH Reverse: 5′ GAAGATGGTGATGGGATTTC 3′ 

### 2.6. HDAC Activity and HDAC Gene Expression

Assays were performed using the HDAC fluorometric activity assay kits from Abcam (ref 156064) as per manufacturer instructions. A total of 100 μg of cell lysate was used for the assays. The HDAC activity of human tumor samples was assessed from tissue samples powdered in liquid nitrogen. Fifty micrograms of tissue powder were dissolved in 300 μL RIPA buffer (250 mM sucrose, 50 mM Tris-Cl, pH 7.5, 25 mM KCl, 5 mM MgCl_2_, 0.2 mM PMSF, 50 mM NaHSO_3_, 45 mM sodium butyrate, 10 mM β-ME, 0.2% TritonX-100) and incubated on ice for 30 min, followed by centrifugation at 5000 rpm for 10 min. The pellet obtained was sonicated (5 s using 30% amplitude) and centrifuged at 15,000 rpm for 30 min. A supernatant equivalent to 1 mg of tissue (5 μL) was used for the HDAC assay. Each tissue sample had its respective blank, i.e., without lysine developer. Fluorescence was estimated using a 96-well plate reader for HDAC. To detect the expression of HDAC genes in CTCL tumor cells, primers for human HDAC sequences (Sigma) were utilized. RNA extraction and cDNA synthesis were performed as described above, and PCR conditions were the following: 4 min/ 95 ºC followed by 40 cycles of 30 s at 95 ºC to denature DNA; 60 ºC/30 s for primer annealing and 30 s at 72 ºC for amplification. Primers used were: 

Forward 5′ AACCTGCCTATGCTGATGCT 3′

Reverse 5′ CAGGCAATTCGTTTGTCAGA 3′

### 2.7. Western Blot

Microtumors were lysed with RIPA buffer (Sigma Aldrich, Baden-Württemberg, Germany) supplemented with phosphatase and protease inhibitors (Roche, Mannheim, Germany). Whole cell lysates were subjected to acrylamide SDS-PAGE using standard procedures, transferred onto a nitrocellulose support membrane (Amersham Protran, GE Healthcare life science, Munich, Germany), and Western blotted. All primary and secondary antibodies were diluted 1:1000 and 1:5000, respectively. The following antibodies were used: anti-α-GAPDH (Santa Cruz Biotechnology, Ref. sc-4772), anti-phospho-ERK Santa Cruz (Cat# sc-7383, RRID:AB_627545, anti-p44/42 MAPK (ERK1/2) (137F5) Cell Signaling Technology (Cat# 4695, RRID:AB_390779), Phospho-Akt (Ser473) Antibody #9271, Akt Antibody #9272, p-Stat5 (5G4): sc-81524, and Stat5 (A-9): sc-74442 from Santa Cruz Biotechnology. Secondary antibodies: P/N 925-32212: RRID AB_2716622, P/N 925-32213: RRID AB_2715510, P/N 926-32213: RRID AB_621848. Signals were visualized and recorded with a Chemi Doc MP Bio Rad system.

### 2.8. Data Analysis and Statistics 

Data processing and statistical analysis were performed using GraphPad Prism 8 Software. Numbers of samples analyzed, and experiments performed are indicated in the Figure Legends. All comparisons were between treated and untreated groups. Welchs’ *t*-test was implemented for comparison of two independent groups. One-way ANOVA analysis with Welch correction was performed for multiple comparison tests. Data are presented as means ± SEM using a one-way ANOVA test with Welch correction used to determine significance (*p* < 0.05) of differences between datasets. Each global mean was compared using a one-way ANOVA test with a statistical significance of *p* < 0.05 (95% confidence interval). *p* values: * < 0.05, ** < 0.01, *** < 0.005, **** < 0.001, ns: not significant.

## 3. Results

### 3.1. Combination of Ruxolitinib with Resminostat Inhibits Transendothelial Migration in Chick Embryo CAM Assay

We first focused on evaluating the transendothelial migration capacity of CTCL cells within the chick embryo model. Notably, individual monotherapies did not yield statistically significant effects on the transendothelial migration of these cells compared to untreated cells. Conversely, marked inhibition was observed when employing a combination scheme with JAKi, ruxolitinib, and HDACi, resminostat. The combined treatment demonstrated a profound inhibitory impact, resulting in a remarkable reduction in CTCL cell transendothelial migration by 75% and 82% (*p* < 0.05) in the respective CTCL engrafted cell lines (MyLa and SeAx) compared to the untreated control group (Figure 1). Moreover, the combined treatment had great inhibitory effect compared to monotherapies as well in both CTCL cells (ruxolitinib: 52% in MyLa, *p* < 0.01; 51% in SeAx, *p* < 0.01; resminostat: 77% in MyLa, *p* < 0.01; 72% in SeAx, *p* < 0.01). These compelling findings highlight the synergistic potential of the JAKi and HDACi combination in significantly impeding the migration of CTCL cells across endothelial barriers, underscoring the therapeutic potential of this dual treatment strategy in mitigating key aspects of CTCL progression within the biological milieu of the chick embryo model.

### 3.2. Combination of Ruxolitinib with Resminostat Blocks Angiogenesis in CTCL Chick Embryo Model

Subsequently, our exploration delves into assessing the impact of HDACi and JAKi on angiogenesis, wherein we effectively showcased the antiangiogenic properties of the combination of resminostat and ruxolitinib in vivo. To substantiate our findings, we conducted a CAM assay to investigate vessel counts post-treatment in comparison to the control. The CAM assay showed a noteworthy reduction in both the overall number and size of blood vessels following the administration of ruxolitinib and/or resminostat. This investigation into chick embryo angiogenesis affirmed that the presence of Ruxolitinib/Resminostat led to diminished vessel numbers and impaired vascular sprouting compared to the vehicle control, attesting to their antiangiogenic characteristics. Specifically, the collaborative action of ruxolitinib and resminostat resulted in a substantial inhibition of angiogƒenesis, significantly diminishing blood vessel formation by 49% and 34% in MyLa (Figure 2A) and SeAx (Figure 2B), respectively (*p* < 0.05). This distinctive combination exhibited pronounced anti-angiogenic effects in the CAM CTCL model, affirming its potential efficacy. Of note, resminostat alone exhibited *a* significant effect only in SeAx engrafted cells (21%, *p* < 0.05) compared to untreated cells, a finding indicating its effectiveness in more advanced disease stages. It must be noted that the combination of ruxolitinib with resminostat had a significant impact on angiogenesis compared to monotherapies (ruxolitinib: 38% in MyLa, *p* < 0.01; 30% in SeAx, *p* < 0.01; resminostat: 26% in MyLa, *p* < 0.01; 28% in SeAx, *p* < 0.05).

Subsequently, the expression levels of angiogenesis-related genes were investigated before and after ruxolitinib with resminostat, and they directly inhibited the expression of the pro-angiogenic VEGFA gene in both engrafted cell lines in chick embryos (61% in MyLa, *p* < 0.01; 79% in SeAx, *p* < 0.005) compared to the untreated group as well as to the monotherapies ruxolitinib: 59% in MyLa, *p* < 0.01; 78% in SeAx, *p* < 0.01; resminostat: 70% in MyLa, *p* < 0.01; 42% in SeAx, *p* < 0.01. Interestingly, resminostat as monotherapy inhibited the VEGFA expression in SeAx engrafted cells, a finding that is important for the efficacy of the drug in advanced stages. On the other hand, the combination of resminostat/ruxolitinib increased the VEGFB anti-angiogenic gene expression in both engrafted cell lines (49% in MyLa, *p* < 0.05; 56% in SeAx, *p* < 0.001) compared to untreated groups (Figure 3B). It is worth noting though that in SeAx engrafted cells, resminostat significantly increased the VEGFB expression compared to untreated cells (26%, *p* < 0.05), and it was more effective than ruxolitinib 24 (%, *p* < 0.05). It is presented that this combination treatment that inhibits pro-angiogenic pathways can lead to a compensatory increase in anti-angiogenic factors.

To further underscore the antiangiogenic prowess of this amalgamation, we gauged hemoglobin levels in MyLa and SeAx cells implanted in the chick embryo model. The combination induced a substantial reduction in hemoglobin levels for both cell lines (MyLa 72%, *p* < 0.01; SeAx 84%, *p* < 0.001) (Figure 4A), with the most significant decrease observed in SeAx cells (Figure 4B) compared to the untreated group as well as to the monotherapies ruxolitinib: 73% in MyLa, *p* < 0.01; 81% in SeAx, *p* < 0.01; resminostat: 61% in MyLa, *p* < 0.01; 68% in SeAx, *p* < 0.01. Notably, resminostat as monotherapy also induced a reduction in hemoglobin levels (36%, *p* < 0.05) in SeAx cells within the chick embryo model. The synergy of resminostat with ruxolitinib demonstrated a potent, antiangiogenic impact, reaching the lowest hemoglobin levels in SeAx cells. This observation implies the potential effectiveness of this combination in addressing advanced stages of angiogenesis.

Moreover, the combination of ruxolitinib with resminostat showed great inhibition regarding tumor mass (MyLa 49%, *p* < 0.01; SeAx 61%, *p* < 0.001), as it is depicted in Figure 5. There was no statistically significant difference between monotherapies and untreated groups in both CTCL engrafted cells. Interestingly, inhibition of tumor mass was greater in combination treatment compared to monotherapies (ruxolitinib: 38% in MyLa, *p* < 0.01; 53% in SeAx, *p* < 0.01; resminostat: 29% in MyLa, *p* < 0.05; 51 in SeAx, *p* < 0.05).

### 3.3. Combination of Ruxolitinib with Resminostat Affects HDAC Activity in CTCL Chick Embryo Model

As far as the HDAC activity is concerned regarding the combination treatment, we first determined the gene expression levels of HDAC, and then we determined the HDAC activity before and after treatment (Figure 6). There was no statistically significant difference in HDAC expression in both engrafted CTCL cells, neither in monotherapies nor in combination treatment compared to untreated groups, a finding indicating that combination treatment and monotherapies did not affect the expression of HDAC at the gene level. On the other hand, the HDAC activity was found to decrease after resminostat and combination treatment at the same levels, a finding suggesting that the combination of resminostat/ruxolitinib had the strongest effect on HDAC activity. 

### 3.4. Major Implicated Pathways Affected by the Combination of Ruxolitinib with Resminostat in CTCL Chick Embryo Model

To further investigate the signal transduction pathways that are activated by single or combinational Resminostat and Ruxolitinib treatment, we performed Western blot analyses in CAM microtumors for key implicated molecules. We investigated the expression of the total AKT protein with the phosphorylated *p*-AKT (Ser473), the total protein ERK with the phosphorylated *p*-ERK (Tyr204), as well as the total STAT5 with the phosphorylated p-STAT5 (Tyr694/699). Normalization of protein levels was achieved using the expression levels of GAPDH.

As is depicted on Figure 7 (Appendix A), in MyLa (Figure 7A), the combination of Resminostat with Ruxolitinib inhibited the phosphorylation of p-AKT by 51% (*p* < 0.01), *p*-ERK by 63% (*p* < 0.01), and *p*-STAT5 by 87% (*p* < 0.005). Interestingly, in SeAx cells (Figure 7B), the monotherapy with ruxolitinib showed a significant inhibition in the phosphorylation of *p*-ERK (32%, *p* < 0.05) compared to untreated groups, whereas the combinational treatment was more effective in the reduction in the phosphorylation of p-AKT by 92% (*p* < 0.001), *p*-ERK by 76% (*p* < 0.01), and *p*-STAT5 by 90% (*p* < 0.001) compared to untreated as well as to monotherapies.

## 4. Discussion

Angiogenesis serves as a pivotal mechanism in the progression of tumor development, facilitating migration and metastasis. Consequently, there is a scientific inquiry aimed at investigating novel therapeutic avenues or identifiable druggable targets with the objective of impeding or restraining angiogenic processes. Given the incurable nature of its prevalent manifestations, Mycosis Fungoides (MFs) and Sézary Syndrome (SS), angiogenesis becomes of particular interest in CTCL. Furthermore, the well-documented involvement of angiogenesis in MF underscores its importance, prompting endeavors to explore therapeutic interventions targeting this process [21,22]. 

Our findings, in accordance with our previous in vivo data [25], elucidated that only the combination of ruxolitinib and resminostat effectively inhibited the transendothelial migration of CTCL cells within the chick embryo model. 

There are limited studies highlighting the antitumor effect of ruxolitinib in other hematological malignancies [13,29] and other solid tumors [24]. In particular, other studies have indicated its effect on HEL leukemia cells, showcasing its ability to inhibit VEGF expression, reduce HIF-1α levels, and impede angiogenesis within this leukemia cell model [13,24]. On the other hand, its administration in vitro led to a dose-dependent inhibition of cell proliferation, accompanied by a simultaneous activation of apoptosis [14]. Another study in a solid tumor of brain glioblastoma investigated the role of ruxolitinib in tumor hypoxia and angiogenesis in vitro following administration of ruxolitinib at various concentrations [30]. The consistent and unaltered expression of HIF-1β regardless of the administered concentration of ruxolitinib was noteworthy. However, the expressions of HIF-1α and VEGF showcased intriguing responses to varying concentrations of ruxolitinib [30]. 

Current research has highlighted the multifaceted effects of an HDAC inhibitor, romidepsin, on cancer cells. Romidepsin’s inhibition of histone deacetylases (HDACs) has been associated with a spectrum of cellular responses, including programmed cell death, cell cycle arrest, induction of apoptosis, stimulation of autophagy, and the inhibition of cell proliferation, angiogenesis, and metastasis. These effects have been observed through the modulation of various signaling pathways within cancer cells. Romidepsin’s broad impact on multiple cellular processes underscores its potential as a therapeutic agent for combating cancer by targeting diverse mechanisms involved in cancer cell survival, proliferation, and metastasis [31]. However, resminostat, an established histone deacetylase (HDAC) inhibitor, has emerged as a promising epigenetic-based therapeutic agent for hepatocellular carcinoma (HCC). In vitro studies have demonstrated its efficacy by exhibiting anti-proliferative and anti-apoptotic properties in HCC cells [32,33]. Furthermore, the combined treatment of resminostat with sorafenib has shown notable potential in mitigating platelet-mediated cancer-promoting effects, specifically within HCC cells. This combination therapy has displayed significant implications for the inhibition of HCC progression by potentially counteracting pro-tumorigenic pathways associated with platelet-mediated mechanisms [34]. 

On the other hand, Mycosis Fungoides and Sézary Syndrome are types of CTCLs that primarily affect the skin and often present as chronic and progressive diseases, with limited curative treatment options available. The understanding that angiogenesis plays a crucial role in the progression of these diseases has led to the investigation of potential therapeutic interventions that could hinder or regulate this process.

Several studies have highlighted the involvement of angiogenesis in CTCL, emphasizing the significance of exploring and developing therapeutic strategies that specifically target this mechanism. By inhibiting or modulating angiogenic processes, researchers aim to impede tumor growth, migration, and metastasis in MF and SS.

The exploration of druggable targets or novel therapeutic avenues to regulate angiogenesis in MF and SS remains an active area of scientific inquiry. Researchers are investigating various approaches, including targeted therapies aimed at specific molecular pathways involved in angiogenesis, anti-angiogenic agents, immunotherapies, and combination treatments that might effectively disrupt the angiogenic processes associated with these lymphomas [25,28].

It is noteworthy to emphasize that despite the promising nature of these investigations, the process of transforming these findings into viable clinical treatments for patients with MF and SS demands an extensive commitment to research, rigorous clinical trials, and thorough validation procedures. This comprehensive approach is essential to ascertaining the safety and efficacy of the potential treatments.

Our findings, in accordance with our previous in vivo data [25] demonstrated that only the combination of ruxolitinib and resminostat inhibited effectively the transendothelial migration and the angiogenesis of CTCL cells within the chick embryo model.

The role of angiogenesis in the context of CTCLs, specifically Mycosis Fungoides (MFs) and Sézary Syndrome (SS), has garnered significant attention within scientific research. Angiogenesis, the formation of new blood vessels, is pivotal in tumor progression and metastasis, and its involvement in these types of lymphomas underscores its relevance and potential as a therapeutic target. 

Our study revealed that the combination of resminostat and ruxolitinib exhibited antiangiogenic effects via the ERK and AKT signaling pathways, aligning with our in vitro data [17]. This combination has shown efficacy in affecting these pathways in various hematological malignancies, playing critical roles in tumor growth, survival, and angiogenesis in CTCL [35,36,37,38,39]. HDAC inhibitors can decrease ERK pathway components’ expression, leading to reduced ERK activation, which diminishes cell proliferation and induces apoptosis [39]. They can also reduce PI3K/AKT pathway activity by downregulating its components, resulting in decreased cell survival and increased apoptosis [39,40]. Moreover, HDAC inhibitors modulate genes involved in angiogenesis and induce endothelial cell apoptosis, disrupting tumor blood vessels.

Ruxolitinib inhibits JAK1/2, reducing STAT protein activation, which affects genes involved in proliferation, survival, and inflammation, interacting with the ERK and PI3K/AKT pathways [41,42,43,44]. Together, resminostat and ruxolitinib more effectively downregulate the ERK and AKT pathways, reducing cell proliferation and enhancing apoptosis [44]. Both pathways promote angiogenesis by regulating VEGF expression [30] By inhibiting these pathways, the combination therapy decreases VEGF production, impairs endothelial cell functions, and disrupts angiogenesis, thereby altering the tumor microenvironment and limiting tumor growth and metastasis. This therapeutic synergy could lead to better clinical outcomes in CTCL compared to monotherapy.

There are limited studies highlighting the antitumor effect of ruxolitinib in other hematological malignancies [24] and other solid tumors [30]. In particular, other studies have indicated its effect on HEL leukemia cells, showcasing its ability to inhibit VEGF expression, reduce HIF-1α levels, and impede angiogenesis within this leukemia cell model [13]. On the other hand, its administration in vitro led to a dose-dependent inhibition of cell proliferation, accompanied by a simultaneous activation of apoptosis [14]. Another study in a solid tumor of brain glioblastoma investigated the role of ruxolitinib in tumor hypoxia and angiogenesis in vitro, following administration of ruxolitinib at various concentrations [30]. The consistent and unaltered expression of HIF-1β regardless of the administered concentration of ruxolitinib was noteworthy. However, the expressions of HIF-1α and VEGF showcased intriguing responses to varying concentrations of ruxolitinib [30]. 

Current research has highlighted the multifaceted effects of an HDAC inhibitor, romidepsin, on cancer cells. Romidepsin’s inhibition of HDACs has been associated with a spectrum of cellular responses, including programmed cell death, cell cycle arrest, induction of apoptosis, stimulation of autophagy, and the inhibition of cell proliferation, angiogenesis, and metastasis. These effects have been observed through the modulation of various signaling pathways within cancer cells. Romidepsin’s broad impact on multiple cellular processes underscores its potential as a therapeutic agent for combating cancer by targeting diverse mechanisms involved in cancer cell survival, proliferation, and metastasis [31]. However, resminostat, an established HDAC inhibitor, has emerged as a promising epigenetic-based therapeutic agent for hepatocellular carcinoma (HCC). In vitro studies have demonstrated its efficacy by exhibiting anti-proliferative and anti-apoptotic properties in HCC cells [32,33]. Furthermore, the combined treatment of resminostat with sorafenib has shown notable potential in mitigating platelet-mediated cancer-promoting effects, specifically within HCC cells. This combination therapy has displayed significant implications for the inhibition of HCC progression by potentially counteracting pro-tumorigenic pathways associated with platelet-mediated mechanisms [34].

On a contrasting note, it is important to highlight that this study investigates for the first time the effects of resminostat in relation to angiogenesis in CTCL. In addition, this distinctive combined therapy showcased an additional noteworthy outcome by effectively hindering angiogenesis, specifically by impeding the formation of new blood vessels. These dual effects not only underscore the potential promise of this combined treatment regimen but also indicate its significant impact in impeding the progression and dissemination of CTCL. This is achieved by concurrently targeting both cell migration and the formation of crucial vasculature essential for the tumor’s sustenance and expansion.

An important observation from our study is the lack of statistically significant effects observed with monotherapies concerning transendothelial migration and angiogenesis. Our investigation revealed that the individual application of therapeutic interventions did not yield statistically significant alterations in the processes of cellular migration across endothelial layers and the formation of new blood vessels. This observation underscores the complexity of these biological phenomena and suggests that a singular therapeutic approach may not be sufficient to induce notable changes in these crucial aspects of cancer progression. The implications point towards the potential necessity for more nuanced strategies, such as combination therapies or targeted interventions, to achieve a more pronounced impact on transendothelial migration and angiogenesis within the studied context. The intricacies of the biological system, the potential limitations of the chosen monotherapies, and the inherent heterogeneity of cancer cells all contribute to the need for further exploration and refinement of therapeutic approaches in order to effectively modulate these specific cellular processes.

Notably, within the framework of SeAx engrafted cells, a compelling observation surfaced wherein resminostat, as a monotherapy, manifested a discernible inhibitory impact on angiogenesis, albeit not as pronounced as the combined approach. This underscores the potency and potential superiority of synergistic combinational therapy over singular treatments, even when certain monotherapies exhibit partial efficacy. The findings underscore the robustness and compelling efficacy of the combined approach in impeding angiogenesis when compared to individual treatments, emphasizing the imperative of employing a multifaceted strategy to achieve optimal therapeutic outcomes in contexts such as CTCL.

## 5. Conclusions

Overall, the profound impact demonstrated by the administration of ruxolitinib and resminostat in attenuating angiogenesis within the in vivo CTCL chick embryo model not only underscores their therapeutic potential but also illuminates a promising avenue for refining treatment paradigms in human subjects. These findings hold immense promise in delineating novel therapeutic strategies centered on angiogenesis inhibition, heralding a new era of more efficacious interventions for individuals afflicted with CTCL. The delineation of such targeted approaches not only augments our understanding of CTCL pathogenesis but also lays a solid foundation for the development of precision-based treatments, thereby fostering enhanced outcomes and improved quality of life for CTCL patients in the foreseeable future.

## Figures and Tables

**Figure 1 cancers-16-03176-f001:**
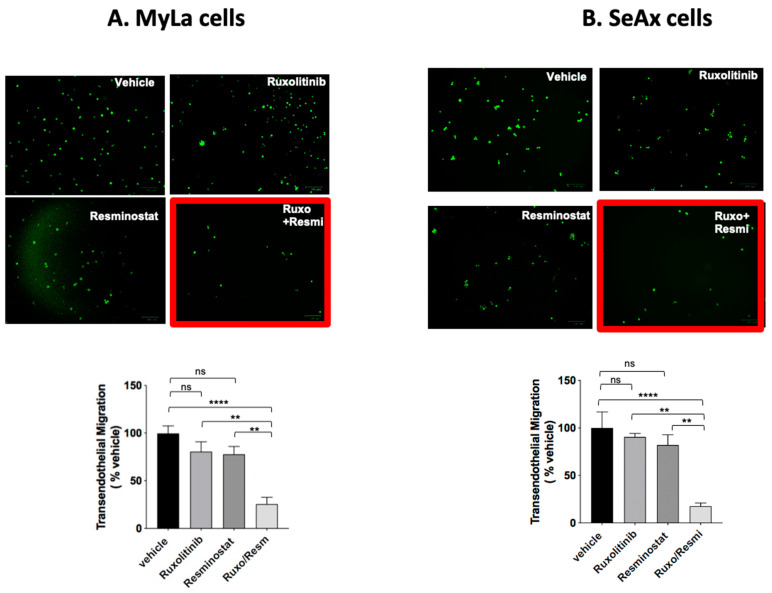
Ruxolitinib/Resminostat inhibited transendothelial migration in MyLa and SeAx engrafted cells in chick embryo model. Values represent mean ± standard error of the mean (SEM) for three independent experiments. Welchs’ *t*-test was implemented for comparison of two independent groups. One-way ANOVA analysis with Welch correction was performed for multiple comparison tests. Significance was defined as *p* ≤ 0.05 and denoted as: ** *p* < 0.01 **** *p* < 0.001, ns: not significant. Analysis was performed using GraphPad Prism 8 software (San Diego, CA, USA).

**Figure 2 cancers-16-03176-f002:**
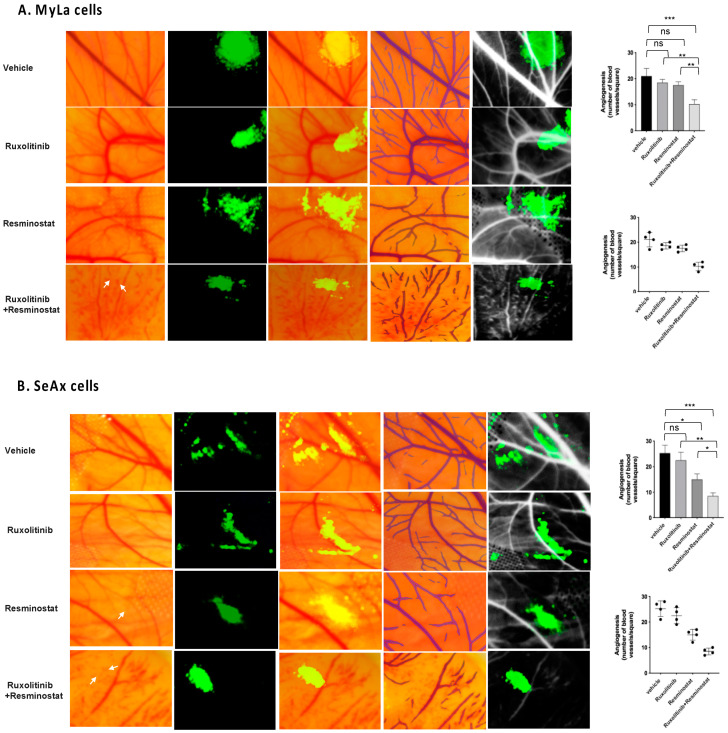
Ruxolitinib/Resminostat blocked angiogenesis in MyLa (**A**) and SeAx (**B**) engrafted cells in chick embryo model. Representative images of (**A**) patterns of vascular branching and IKOSA blood vessels map (**left panel**) and its quantification (**right panel**) in grafted CTCL microtumors into the CAM treated as indicated. Arrows in the image indicate the effect of drugs on blood vessel density and integrity. Combination of ruxolitinib and resminostat resulted in the development of thin and collapsed blood vessels with reduced size lumens compared to control. Values represent mean ± SEM for four independent experiments, each employing 10–12 embryos per treatment variant. * *p* < 0.05, ** *p* < 0.01, *** *p* < 0.005, ns: not significant by one-way ANOVA test.

**Figure 3 cancers-16-03176-f003:**
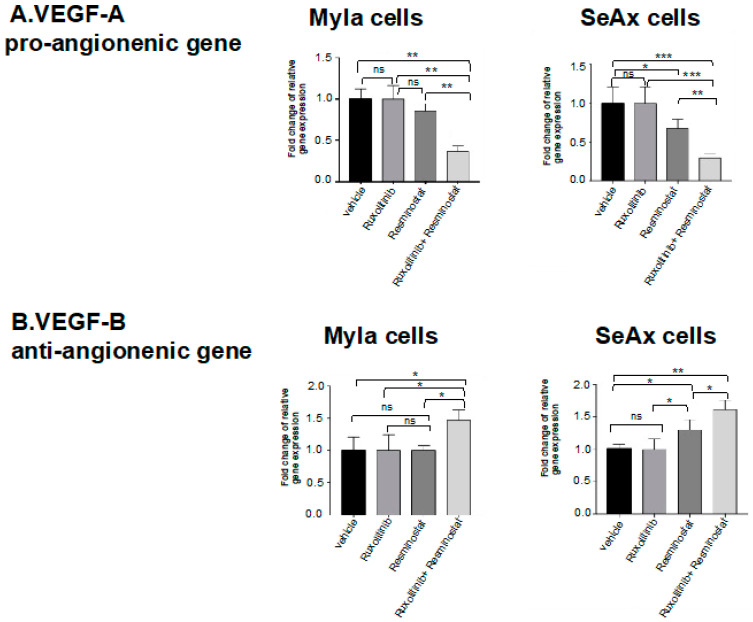
Expression levels of angiogenesis-related genes in MyLa/SeAx engrafted cells in chick embryo model after treatment with resminostat and/or ruxolitinib. Combination treatment decreased VEGFA (**A**) and increased VEGFB (**B**) in CTCL cells engrafted in chich embryos compared to vehicle. The bars are the means determined in three (*n* = 3) independent experiments using 10–12 embryos per variant. * *p* < 0.05, ** *p* < 0.01, *** *p* < 0.005, ns: not significant by one-way ANOVA test.

**Figure 4 cancers-16-03176-f004:**
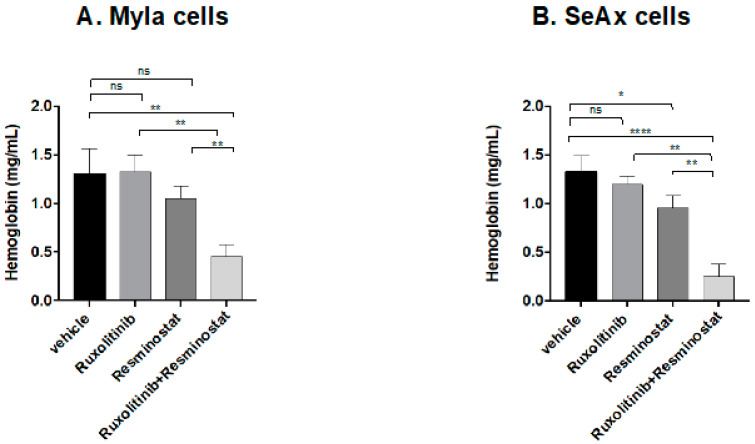
Ruxolitinib/Resminostat decreased hemoglobin levels in MyLa (**A**) and SeAx (**B**) engrafted cells in chick embryo model compared to vehicle. The bars are the means determined in three (*n* = 3) independent experiments using 10–12 embryos per variant. * *p* < 0.05, ** *p* < 0.01, **** *p* < 0.001 by one-way ANOVA test.

**Figure 5 cancers-16-03176-f005:**
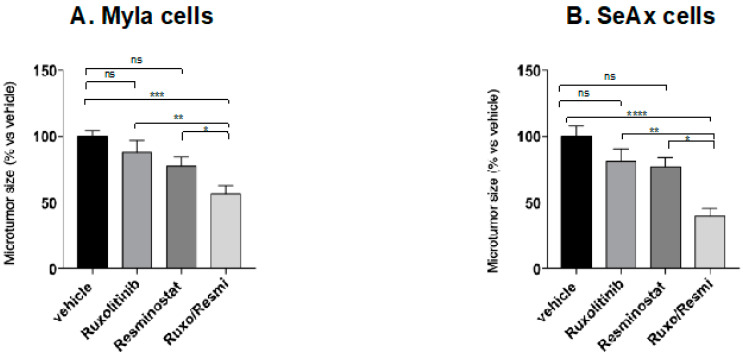
Ruxolitinib/Resminostat decreased tumor mass in MyLa (**A**) and SeAx (**B**) engrafted cells in chick embryo model compared to vehicle. Data show mean ± SEM from three (*n* = 3) independent experiments, each employing 12–14 embryos per treatment variant. * *p* < 0.05, ** *p* < 0.01, *** *p* < 0.005, **** *p* < 0.001 by one-way ANOVA test.

**Figure 6 cancers-16-03176-f006:**
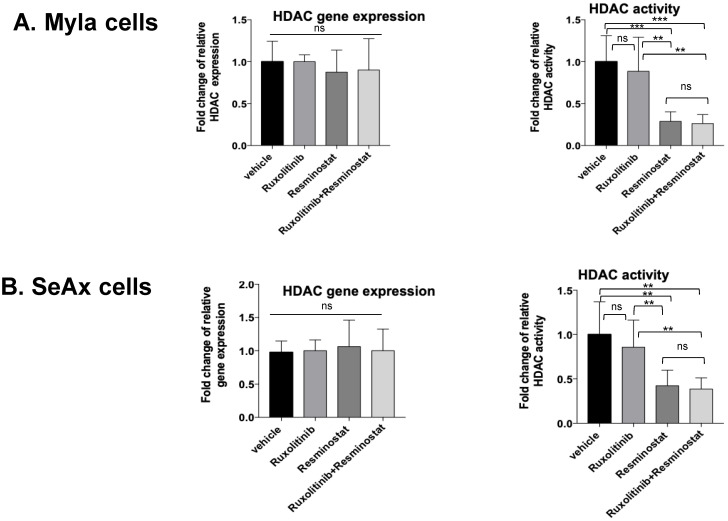
*HDAC* gene expression and HDAC activity in MyLa (**A**) and SeAx (**B**) engrafted cells in chick embryo model. Combination of Ruxolitinib/Resminostat and resminostat alone decreased HDAC activity levels in MyLa and SeAx engrafted cells in chick embryo model. Data show mean ± SEM from three (*n* = 3) independent experiments, each employing 10–12 embryos per treatment variant. ** *p* < 0.01, *** *p* < 0.005 , ns: not significant by one-way ANOVA test.

**Figure 7 cancers-16-03176-f007:**
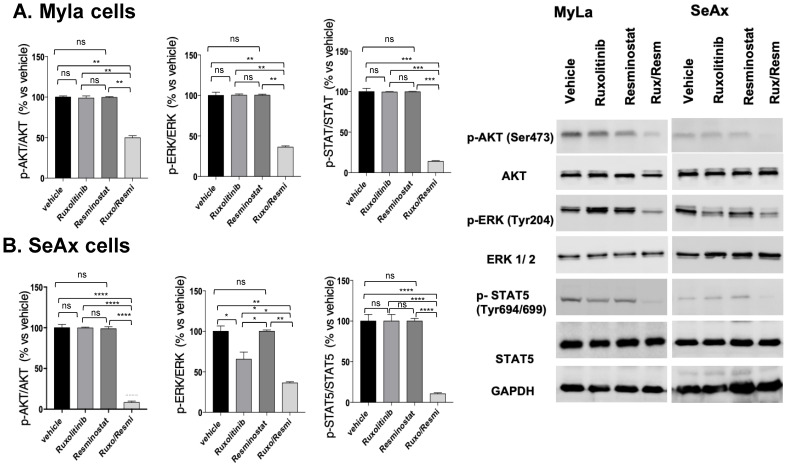
Western blot analyses in CTCL CAM onplant microtumors for key implicated pathways. Portions of microtumors were lysed and analyzed in MyLa (A) and SeAx (B) CAM onplant microtumors for the activation levels p-AKT (Ser473), p-ERK (Tyr 204), p-STAT5 (Tyr694/699), and GADPH in . Data show mean ± SEM from three (*n* = 3) independent experiments. *p* values: * < 0.05, ** < 0.01, *** < 0.005, **** < 0.001 by one-way ANOVA test.

## Data Availability

The data presented in this study are available in this article.

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
