# Peer review of "Combination of JAKi and HDACi Exerts Antiangiogenic Potential in Cutaneous T-Cell Lymphoma"

_cancers, 2024, doi:10.3390/cancers16183176_

Round 1
Reviewer 1 Report (Previous Reviewer 2)
Comments and Suggestions for Authors
In the revised version, the authors provided more evidences to support their conclusions and also made corrections according to the review. My concerns were addressed except statistical analysis. In this study, there were four groups which should be analysis with One-way ANOVA method. The Student's t-test is a statistical test used to compare the means of two groups which is inappropriate in this study. The authors should show the comparisons between vehicle and Ruxolitinib or Resminostat or combination of Ruxo/Resm respectively, as well as the comparisons between combination of Ruxo/Resm and Ruxolitinib or Resminostat. If there is no significance, then mark “ns”.
Author Response
Reviewer 1:
In the revised version, the authors provided more evidences to support their conclusions and also made corrections according to the review. My concerns were addressed except statistical analysis. In this study, there were four groups which should be analysis with One-way ANOVA method. The Student's t-test is a statistical test used to compare the means of two groups which is inappropriate in this study. The authors should show the comparisons between vehicle and Ruxolitinib or Resminostat or combination of Ruxo/Resm respectively, as well as the comparisons between combination of Ruxo/Resm and Ruxolitinib or Resminostat. If there is no significance, then mark “ns”.
Response:
We appreciate your feedback. We conducted a One-way ANOVA analysis, and where there was no significance, we indicated this with the mark “ns.” All changes suggested by the reviewer have been accepted, and we have also highlighted the updated results in green in the revised manuscript. The updated version of the manuscript has been downloaded from the platform "Manuscript for Revisions" manuscript.v2.doc.

This manuscript is a resubmission of an earlier submission. The following is a list of the peer review reports and author responses from that submission.
Round 1
Reviewer 1 Report
Comments and Suggestions for Authors
The authors studied the effect of the combination of Ruxolitinib with Resminostat to CTCL cell lines in ckick embryo model. The combination therapy showed anti-angiogenesis and inhibited tumor migration. However, they didn’t show whether the combination therapy could decrease tumor volume or not in vivo. Moreover, they should show anti-angiogenesis in treated tumor mass.
Reviewer 2 Report
Comments and Suggestions for Authors
Comments: In this work, the authors investigated the effects of Ruxolitinib (JAKi) in combination with Resminostat (HDACi) treatment in transendothelial migration of CTCL cells (MyLa and SeAx) by using a transwell-based transendothelial migration assay and CTCL chick embryo Chorioallantoic membrane (CAM) model. They found that the combination of Ruxolitinib with Resminostat inhibited CTCL cell transendothelial migration and blocked angiogenesis in both MyLa and SeAx cells. Although this study has certain clinical significance, the authors need to provide more evidence to support their conclusions. The statistical methods used in this study was inappropriate. The design of the study was not adequate and reasonable. The manuscript was not presented clearly enough.
#1. What are the changes in the expression levels of angiogenesis-related genes after the combination of Ruxolitinib and Resminostat treatment?
#2. What is the molecular mechanism of the combined treatment of these two drugs in inhibiting angiogenesis?
#3. Please verify the expression level of the corresponding protein treated by these drugs, for example, expression level of HDAC (Histone Deacetylase) family after Resminostat treatment.
#4. Student’s t-test is not suitable for analysis of differences between more than two groups.
#5. Please clearly label statistical differences between which two groups. If there is no statistical significance between two groups, please mark “ns”.
#6. Please label “A” or ”B” in all figures.
#7. Please clarify the timing and concentration of drug treatment in Materials and Methods. And explain why there was no testing of different gradient concentrations and different time treatments of these drugs.
Comments on the Quality of English Language
The manuscript was not presented clearly enough.